# Unveiling the role of interleukin-13 in liver fibrosis of chronic hepatitis B patients: Development of a predictive model

**Muhammad Begawan Bestari**[ID]<sup>☯</sup>*, **Muhammad Palar Wijaya**[ID]<sup>☯</sup>, **Dolvy Girawan**<sup>☯</sup>, **Nenny Agustanti**<sup>☯</sup>, **Eka Surya Nugraha**[ID]<sup>☯</sup>

Division of Gastroenterohepatology, Department of Internal Medicine, Faculty of Medicine Universitas Padjadjaran, Hasan Sadikin General Hospital, Bandung, Indonesia

☯ These authors contributed equally to this work.
* begawan@unpad.ac.id

## Abstract

### Background

Developing a non-invasive model is essential for assessing liver stiffness in conditions without transient elastography, which will determine further management in chronic hepatitis B (CHB) patients.

### Objectives

This study aims to evaluate the interleukin-13 role in liver fibrosis and develop a new predictive model that includes interleukin-13 and standard data such as age and platelet count.

### Methods

Patients were recruited from Hasan Sadikin General Hospital's CHB registry from October 2021 to January 2022. Patients underwent demographic data collection, complete blood count, interleukin-13, and transient elastography examinations on the same day. The platelet count variable was listed in $\times 10^9$/ dL, and the interleukin-13 in pg/mL. Interleukin-13 values were categorized as positive for IL-13 with a cut-off of 5 pg/mL (ILcut5). The liver stiffness measurement was inverted to obtain a normal distribution and became inverseLSM as the model's outcome. The prediction model was formed through multiple linear regression analysis.

### Results and Discussion

The number of patients studied was 88. A prediction model for inverseLSM was formulated, had an $R^2$ of 0.37, and consisted of age, platelet count, and ILcut5 with

**Data availability statement:** The raw data are available from Figshare: https://doi.org/10.6084/m9.figshare.28388003.v1 The TRIPOD Checklist for the development of a prediction model as the reporting guideline used is available from Figshare: https://doi.org/10.6084/m9.figshare.28425002.v1 The statistical analysis step is available from Figshare: https://doi.org/10.6084/m9.figshare.28425038.v1.

**Funding:** This publication charge is funded by Universitas Padjadjaran (Unpad) through the Indonesian Endowment Fund for Education (LPDP) on behalf of the Indonesian Ministry of Higher Education, Science and Technology and managed under the EQUITY Program (Contract No. 4303/B3/DT.03.08/2025 and 3927/UN6.RKT/HK.07.00/2025).

**Competing interests:** The authors declare that they have no competing interests. Specifically, within the five years preceding the conduct of this research and the preparation of this manuscript, the authors have not had any financial relationships or activities that could be perceived as influencing the work, including but not limited to employment, consultancy, board membership, stock or share ownership, patent applications (pending or issued), research grants, travel grants, honoraria, or gifts from any commercial or non-commercial entities. The authors also declare no non-financial competing interests, including acting as expert witnesses, membership in government or advisory boards, affiliations with advocacy or lobbying organizations, relationships with funding bodies or educational companies, personal or professional relationships with individuals involved in the submission or evaluation of this manuscript, or personal convictions that could reasonably be perceived as influencing the objectivity of the research. Interests outside the five-year time frame that could reasonably be perceived as competing are also absent.

correlation coefficients of -0.221, 0.326, and −0.288, respectively. The model had no autocorrelation, multicollinearity, or significant outliers, a normal distribution appearance, and met the homoscedastic criterion. Elevated IL-13, decreased platelet count, and aging are linked to increased liver stiffness.

## Introduction

Chronic hepatitis B (CHB) occurs when hepatitis B surface antigen (HBsAg) is present in blood serum for over six months. Age of exposure is crucial for determining the risk of developing a chronic infection. [1] People who are HBsAg-positive need to be checked regularly to monitor indications for treatment. [2] Some necessary things that need attention include levels of Hepatitis B Envelope Antigen (HBeAg), aminotransferase, Hepatitis B Virus (HBV) DNA, and the importance of assessing the stage of liver fibrosis through several non-invasive tests (NITs). [1] CHB treatment is recommended for people with evidence of significant fibrosis/ cirrhosis, and it is explicitly stated to assess it through the aspartate aminotransferase (AST) to Platelet Ratio Index (APRI) score or transient elastography (TE) value. TE services require trained skills and significant effort in procuring the equipment [1].

Through multivariate analysis of various primary patient data, prediction models were created to find the best model for determining the significance of liver fibrosis. [3,4] A systematic multivariate analysis formed a simple model involving platelet count and AST levels. [3] After a multivariate analysis, platelets, age, AST, and alanine aminotransferase (ALT) formed a Fibrosis-4 Index (FIB-4) model to predict liver fibrosis [4].

The patient's age is the primary personal data that is always considered in the differential diagnosis of a disease. The likelihood of developing chronic liver disease (CLD) rises with age. Apart from that, old age is the main factor for complications and puts patients in dangerous conditions. [5] Platelet count is an accessible variable from a blood test. It is an uncomplicated marker for assessing the extent of liver fibrosis in individuals with CHB. Its diagnostic ability as a single marker can compete with FIB-4 and APRI. [6]

Interleukin-13 (IL-13) is a messenger substance in the immune system. [7] This compound plays a specific role in wound healing and steatohepatitis and has been identified in cohorts with CHB. [8] Multiple studies have demonstrated that IL-13 can influence fibrosis progression through increased collagen production, regulation of proteolytic enzymes, and activation of profibrotic signaling pathways. [8–12] To our knowledge, IL-13 has not been previously established as independently associated with liver fibrosis. Previous studies investigating the association between IL-13 and liver fibrosis have yielded conflicting results. [8–12] Therefore, further confirmation using a more comprehensive analytical approach, multiple linear regression analysis, is warranted. This study aims to evaluate the role of IL-13 in liver fibrosis and to develop a new predictive model that incorporates IL-13, age, and platelet count.

## Materials and methods

### Participants and data

This study employed a cross-sectional design with a population of CHB patients. Hasan Sadikin General Hospital is one of Indonesia's premier national referral hospitals, providing the setting for patient selection. Patients were selected from Hasan Sadikin General Hospital's CHB registry. The CHB registry has been functioning since October 2021 and includes patients over 18 with positive HBsAg results for at least six months. Patients with Hepatitis C, autoimmune liver disease, HIV, type 2 diabetes mellitus, heart disease, chronic kidney disease, pulmonary tuberculosis, cancer, history of alcohol use (20 grams per day), pregnant or breastfeeding women, body mass index (BMI) >27 kg/m$^2$, hemoglobin <5 g/dL, pulmonary fibrosis, and chronic pancreatitis are not included in this registry. The participants for this study were drawn from the same registry as the Mac-2-Binding Protein Glycosylation Isomer (M2BPGi) study, which has been previously described in detail. [13] Blood samples and FibroScan® measurements were obtained from patients prior to the initiation of hepatitis B therapy.

The number of patients studied was 88 from 21 October 2021 until 21 January 2022. According to the rule of thumb method, 5−10 subjects per independent variable were enough for multivariate linear regression analysis. Thus, 15−30 subjects are needed for three independent variables. Using the α of 5%, β of 20%, and R$^2$ of 0.25, with three independent variables, the minimum sample size is 37 subjects. A sample size of 88, if included in the equation to assess the minimum sample size needed to create a prediction model, also met the criteria. [14] Following a systematic review and adherence to research and professional ethics guidelines, the Research Ethics Committee of Dr. Hasan Sadikin General Hospital in Bandung approved this research (LB.02.01/X.6.5/299/2021). All patients received research information and completed a written consent form. Patients would undergo demographic data collection, complete blood count, IL-13 serum examinations, and TE on the same day. All laboratory examinations were carried out in the clinical pathology laboratory of Hasan Sadikin General Hospital.

### Predictors

Three independent variables were used as predictors. The age variable was calculated from the examination date minus the date of birth. Numbers were rounded down so that the age variable could be collected without decimal places. The platelet count variable was listed in new units (×10$^9$/ dL) to assess changes per 10,000 platelets in international units (×10$^9$/ L). Age and platelet count were analyzed as numerical variables.

IL-13 was measured with the IL-13 ELISA Kit (Elabscience Biotechnology Co., Ltd., Catalog number: E-EL-H0104, Lot: CV05V22Z9069), which uses the Sandwich-ELISA principle method. The serum was mixed with the antibodies specific to Human IL-13, biotinylated detection antibodies specific to Human IL-13, and Avidin-Horseradish Peroxidase (HRP) conjugate. Furthermore, the optical density (OD) was measured spectrophotometrically at 450 nm ± 2 nm to assess the concentration of human IL-13. IL-13 values were analyzed as the nominal variable.

### Outcomes

The outcome variable in this research is Liver Stiffness Measurement (LSM), which was obtained from TE results expressed in kPa units. This examination was performed with the FibroScan® 502 Series F00734 (Echosen, Paris, France) with the M probe. The TE method required a minimum of 10 valid readings, ensuring a 60% success rate and an interquartile range below 30% of the median values obtained. A gastroenterohepatologist performed the TE examination on the same day as other examinations at the Hasan Sadikin General Hospital. From now on, the LSM variable underwent data transformation to obtain a normal distribution called the inverseLSM variable.

$$inverseLSM = \frac{1}{LSM + 1}$$

## Statistical analysis

Initially, a normality test was carried out to see the distribution of each variable used. The IL-13 variable was made into categorical data (ILcut5) based on the cut-off value previously obtained through receiver operating characteristic (ROC) curve analysis and the implementation of the Youden index calculation. Bivariate analysis was performed on inverseLSM as the dependent variable and age, platelets, and ILcut5 as independent variables. Multiple linear regression (MLR) analysis was done with the three predictor variables (age, platelet count, and IL-13) at once to see their role in determining the inverseLSM value.

We checked all assumptions during the multiple linear regression (MLR) analysis to validate the results. The Durbin-Watson statistics were used to assess the independence of observations within the model. Scatterplots were analyzed to examine the relationships between the predictors and the outcomes. The homoscedasticity of the model was evaluated through scatterplots and the Breusch-Pagan test. We detected multicollinearity using the bivariate correlation test, variance inflation factor (VIF), and tolerance metrics derived from the collinearity statistics. Outliers were identified using Cook's distance, standardized residuals, and scatterplots. The distribution of residuals was assessed through the P-P plot, normality tests, and standardized residuals. SPSS version 20 (IBM Corp. 2011. Armonk, NY, USA, RRID: SCR_00286) was used for statistical analysis, and a p-value <0.05 was statistically significant.

## Results

### Subject characteristics

From October 2021 until January 2022, 102 patients were found with complete data. Every patient who was willing to participate would be included in the study. Then, 88 patients were analyzed. The overall characteristics of the 88 participants are explained in Table 1.

### Model development

Through ROC curve analysis followed by computing the Youden index to maximize sensitivity and specificity, the cut-off for IL-13 was obtained at 5 pg/ mL in the diagnosing HR group. Any IL-13 values greater than or equal to 5 pg/ mL will be categorized as positive for the IL-13 examination (ILcut5 = 1). The outcome variable became normally distributed after transforming the LSM variable into the inverseLSM. The normality test results after the transformation of the outcome variable and the selection of the predictor variable type are listed in Table 2.

Age, platelet count, and inverseLSM were normally distributed, so they were valid for inclusion in multivariate linear regression analysis as numerical data (Table 2). Before developing the prediction model, the correlations between variables were analyzed (Table 3).

**Table 1. Overall characteristics of 88 study participants.**

| Characteristics | Value |
|---|---|
| Total number of patients (n) | n = 88 |
| Age (years)‡ | 43.85 (±14.077) |
| Sex (male/ female)* | 50/38 |
| Platelets (×10⁹/dL)‡ | 23.534 (±7.855) |
| IL-13 (pg/mL)† | 3.485 (0.1-16.79) |
| LSM (kPa)† | 22.45 (3.4-70.6) |

* Value is given as a proportion.

† Values are the medians with ranges in parentheses.

‡ Values are the means with standard deviations in parents.

**Table 2. The normality test results of each variable.**

| Characteristics | p-value |
|---|---|
| Age (years)‡ | 0.068 |
| Platelets (×10⁹/dL)‡ | 0.174 |
| inverseLSM‡ | 0.133 |
| ILcut5 (positive/ negative)* | 23/65 |

‡The p-value is based on the saphiro-wilk test, * Value is given as a proportion; ILcut5, IL-13 divided categorically by cut-off ≥5 pg/mL.

**Table 3. Bivariate correlation results between dependent and independent variables.**

| Variable Independent | Variable Dependent | r | p-value |
|---|---|---|---|
| Age | Inverse LSM | −0.427‡ | <0.001 |
| Platelets | Inverse LSM | 0.470‡ | <0.001 |
| ILcut5 | Inverse LSM | −0.426† | <0.001 |

r, Coefficient correlations; ILcut5, IL-13 divided categorically by cut-off ≥5 pg/ mL.

‡ Coefficient correlations based on the Pearson test.

† Coefficient correlations based on the point biserial.

Based on Table 3, there was a correlation between each variable. The highest correlation between variables did not exceed 0.7, indirectly indicating no multicollinearity if further analysis was conducted. Moreover, each independent variable had a p-value of <0.25, which can be included in the subsequent multivariate analysis. MLR analysis was performed using the age and platelet count as continuous variables, and the IL-13 was applied as a nominal variable using the cut-off value obtained, the ILcut5. Through MLR analysis, a prediction model for inverseLSM was formulated. (Table 4).

The model had an R-squared of 0.37, which means that this model covered 37% of the variance in the inverseLSM results. Significant results were obtained from ANOVA analysis to predict inverseLSM (p-value < 0.001). The Durbin-Watsons test result was 1.686. Two conclusions were obtained from the analysis of the Durbin-Watson table: there was no negative autocorrelation, but it was still inconclusive for positive autocorrelation. Plotting the dependent variable against the regression standardized predicted value gave a linear impression. The scatterplot involving standardized predicted values and standardized residual data was equal in the four quadrants. It can be concluded that the residuals did not increase with the increment of the independent variable. A computerized Breusch Pagan analysis yielded a p-value greater than 0.05. The Cook's distance was between 0.00 and 0.062, and the standardized residual values were (−2.5) to (+2.45). When plotted on a P-P plot, the standardized residual values sufficiently aligned to create a straight diagonal line,

**Table 4. Multivariate analysis to identify independent predictors of inverseLSM value.**

| Model | | Unstandardized Coefficients | | Standardized Coefficients | t | Sig. | Collinearity Statistics | |
|---|---|---|---|---|---|---|---|---|
| | | B | Std. Error | Beta | | | Tolerance | VIF |
| 1 | (Constant) | 0.107 | 0.025 | | 4.291 | 0.000 | | |
| | Age (Years) | −0.001 | 0.000 | −0.221 | −2.305 | 0.024 | 0.815 | 1.226 |
| | Plat (10⁹/dL) | 0.002 | 0.001 | 0.326 | 3.479 | 0.001 | 0.853 | 1.173 |
| | ILcut5 | −0.034 | 0.011 | −0.288 | −3.147 | 0.002 | 0.896 | 1.116 |

Dependent Variable: inverseLSM. $R^2$= 0.37 (N= 88, p-value < 0.001).

ILcut5, IL-13 divided categorically by cut-off ≥5 pg/ mL; t, t value from two-sided t test; Sig., the two-sided p-value for each B-coefficient; VIF, Variance Inflation Factor.

indicating that both the residuals and standardized residuals followed a normal distribution. Analysis of the tolerance value was always greater than 0.1, and the VIF values were always smaller than 10, sufficient to prove that no multicollinearity was found in our model. From the MLR analysis, a predictive equation for inverseLSM was produced as follows:

$$\frac{1}{(LSM + 1)} = 0.107 - (0.001\ Age) + (0.002\ Platelets\ (10^9/dL) - (0.034\ ILcut5)$$

ILcut5 = 1, if IL-13 ≥ 5 pg/ mL

The model had no autocorrelation, no multicollinearity, no significant outliers, had the appearance of a normal distribution, and met the homoscedastic criteria. In exploratory analyses, we observed statistically significant interactions between IL-13 and age (p = 0.02) and between IL-13 and platelet count (p = 0.022). However, each interaction explained only a small proportion of the overall variance ($R^2$ = 0.06) (S1 Table).

## Discussion

Our study showed a significant correlation between age, platelet, and IL-13 to LSM values. (Table 4). Aging will increase LSM values, assuming that young age is still in the immunotolerant phase and significant fibrosis has not been found. There was a significant difference in LSM values in each age group. [15] The increase in LSM will be accompanied by a decrease in platelets in this study. Platelet count was found to have a correlation coefficient of −0.495 with LSM from a Spearman test.(15) We obtained a correlation coefficient 0.326 for platelet count in an inverseLSM predictive model from multiple linear regression analysis. Another study also supports this possibility; the platelet count decreased with each increase in the METAVIR score from F0 to F4 in patients with CHB. [6] The rise in IL-13 levels also increased LSM values. IL-13 later induces connective tissue growth factor (CTGF) in HSCs. [16] Another effect of IL-13 release is to cause HSCs proliferation and simultaneously increase the expression of TGF-β and platelet-derived growth factor (PDGF), which play a significant role in the chronic course of liver fibrosis [17].

Most previous studies analyzed predictors for liver fibrosis using logistic regression analysis. Therefore, their analyses could not be directly compared to our results, which resulted from a multivariate linear regression analysis. The coefficient correlation from the highest to lowest were defined by platelet count, IL-13, and Age with 0.326, −0.288, and −0.221, respectively, in predicting inverseLSM. In an analysis of various independent variables in patients with CLD, age was only proven to be a predictor of LSM > 7 kPa in the univariate logistic regression analysis with OR: 1.05 (95%CI 1.03–1.07). Platelet count showed more vital ability in a multivariate logistic regression analysis with OR: 0.981 (95%CI 0.972–0.99). [18] Similar results were seen in CHB-only patients; age was only significant in univariate logistic regression predicting Batts and Ludwig score ≥F2. The Platelet count had a p-value of 0.01 in the multivariate logistic regression with OR: 0.994 (95%CI 0.99–0.999). [19] In predicting Ishak score ≥F2, Platelet count was the only independent predictor in multivariate logistic regression with OR: 0.9 (95%CI 0.9–1). Meanwhile, the age variable was insignificant from univariate or multivariate logistic regression. [20] Although no one used the MLR analysis method as we did, there were quite different results from ours, which found that platelet count and age could predict LSM synchronously.

The role of IL-13 in hepatitis B was recently discovered in a Malaysian cohort. Plasma IL-13 was independently associated with increasing hepatic steatosis as assessed by the controlled attenuation parameter (CAP) examination. In that study, IL-13 was unrelated to LSM outcome scores. [8] Our optimal IL-13 cut-off was identified using the Youden index, which selects the threshold that maximizes the combined sensitivity and specificity on the ROC curve. [21] This approach was applied to our internal dataset to derive a cut-off that provides a balanced discrimination performance while maintaining clinical interpretability within the characteristics of the studied cohort. A positive for ILcut5 was associated with a decrease in inverseLSM of 0.288; in other words, an increase in IL-13 was associated with an increase in LSM. (Table 4) Recent studies have shown that fibrogenesis in CLD patients depends on the effects of IL-13. [9] In patients with HCV,

IL-13 expression decreased proteinase and collagenase expression, thus correlated with fibrosis stage. [10] HIV-HCV coinfected patients showed a higher IL-13 value in severe liver fibrosis condition determined by FIB-4 > 3.25. [11] Through receptor, gene, and protein expression analysis, IL-13 induces type I collagen production in IL-13 receptor subunit α2 – positive (IL-13Rα2-positive) HSC via the transforming growth factor-β (TGF-β) pathway. [12] Another related receptor is IL-13 receptor subunit α1 (IL-13Rα1) for Janus kinase/signal transducer and activator of transcription-6 (JAK-STAT-6) signal transduction which causes the production of Eotaxin-1 (CCL11), an eosinophil chemotactic protein. CCL11 was always independently associated with LSM in all models through the MLR analysis [8].

Potential interaction effects were explored by incorporating interaction terms between IL-13 and clinically relevant variables. Although both the IL-13–age interaction and the IL-13–platelet count interaction reached statistical significance in our study, their modest effect sizes, together with the limited sample size, warrant cautious interpretation. Further studies in larger population are needed to clarify whether these interactions represent meaningful biological or clinical effect modification.

Single, indirect liver fibrosis biomarkers were significant in determining liver fibrosis status; notable examples include AST, ALT, Albumin, alkaline phosphatase (ALP), gamma-glutamyl transferase (GGT), bilirubin, total leukocytes, and INR. [19,22] Our analysis did not incorporate these potential predictors when constructing the liver fibrosis model, which may explain the low R-squared value obtained. Liver fibrosis is a series of complex processes based on chronic injury to the liver, which will later trigger oxidative stress and, if left untreated, will cause chronic inflammation. This process will cause wound healing in the liver, which, if repeated, will cause liver fibrosis [23].

The primary and assumptions tests indicated that our overall analysis was unbiased, consistent, and efficient in estimating the regression coefficients and the standard errors. However, this study has several limitations. Firstly, the participants were exclusively CHB patients, meaning our findings only apply to this group. Consequently, external validation studies are necessary to evaluate the applicability of our model to other CLD groups. Despite this limitation, developing a predictive model for liver fibrosis is essential to supporting health services without TE and developing the treasure of science. Second, because some previous studies showed that the role of IL-13 was only related to liver steatosis, further confirmation is needed to investigate its role in liver fibrosis. Discrepancies between our findings and prior reports may be attributable to biological variability across populations, differences in patient characteristics, variations in IL-13 measurement techniques, and heterogeneity in analytical approaches. In addition, the IL-13 cut-off obtained was derived from our internal cohort and may be influenced by population characteristics and assay differences, while dichotomizing a continuous variable may reduce statistical power and lead to population-specific thresholds. Although our sample size met the minimum requirements for model development, the relatively small number of participants may limit model stability and reduce the ability to detect interaction effects. Therefore, both the proposed IL-13 cut-off and the predictive model require validation in larger and independent populations before broader application.

## Conclusion

In conclusion, it has been established that IL-13 is independently associated with liver fibrosis, making it a significant part of the predictive model for liver fibrosis in CHB. Higher IL-13 levels, reduction in platelet count, and advanced age appear to play a role in enhancing liver stiffness. The application of MLR analysis in this study provides a very important methodological basis for future research. This study is anticipated to broaden scientific insights and support the advancement and acceleration of CHB management.

## Supporting information

**S1 Table. Exploratory Interaction Analyses in Multiple Linear Regression Models.** Available at https://doi.org/10.6084/m9.figshare.31479409.
(DOCX)

**S2 Table. The Raw Study Data.** Available at https://doi.org/10.6084/m9.figshare.28388003.v1.
(XLSX)

**S3 Table. The TRIPOD Checklist for the Development of A Prediction Model.** Available at https://doi.org/10.6084/m9.figshare.28425002.v1.
(PDF)

**S4 Table. The Statistical Analysis Step.** Available at https://doi.org/10.6084/m9.figshare.28425038.v1.
(DOCX)

**S1 Fig. The Supporting Plot for the Model.** Available at https://doi.org/10.6084/m9.figshare.31479883.
(DOCX)

## Acknowledgments

We sincerely thank all staff of the gastroenterohepatology services and the department of clinical pathology at Hasan Sadikin General Hospital for their invaluable support in executing this research.

## Author contributions

**Conceptualization:** Muhammad Begawan Bestari, Dolvy Girawan.

**Data curation:** Muhammad Begawan Bestari, Muhammad Palar Wijaya.

**Formal analysis:** Muhammad Begawan Bestari, Muhammad Palar Wijaya, Dolvy Girawan, Eka Surya Nugraha.

**Investigation:** Muhammad Begawan Bestari.

**Methodology:** Muhammad Begawan Bestari, Nenny Agustanti.

**Project administration:** Muhammad Palar Wijaya.

**Software:** Muhammad Palar Wijaya.

**Supervision:** Muhammad Begawan Bestari, Dolvy Girawan, Nenny Agustanti, Eka Surya Nugraha.

**Validation:** Muhammad Begawan Bestari, Muhammad Palar Wijaya, Dolvy Girawan, Nenny Agustanti.

**Visualization:** Muhammad Begawan Bestari, Dolvy Girawan, Nenny Agustanti, Eka Surya Nugraha.

**Writing – original draft:** Muhammad Palar Wijaya.

**Writing – review & editing:** Muhammad Begawan Bestari, Muhammad Palar Wijaya, Dolvy Girawan, Nenny Agustanti, Eka Surya Nugraha.

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
