## [Decision Letter · Decision Letter 0]

20 Oct 2025

Dear Dr. Bestari,

Thank you for submitting your manuscript to PLOS ONE. After careful consideration, we feel that it has merit but does not fully meet PLOS ONE’s publication criteria as it currently stands. Therefore, we invite you to submit a revised version of the manuscript that addresses the points raised during the review process.

Please respond to reviewers' comments individually.

We look forward to receiving your revised manuscript.

Kind regards,

Xiaosheng Tan

Academic Editor

PLOS ONE

Journal Requirements:

“NO authors have competing interests”

Reviewers' comments:

Reviewer's Responses to Questions

**Comments to the Author**

1. Is the manuscript technically sound, and do the data support the conclusions?

Reviewer #1: Yes

Reviewer #2: Yes

Reviewer #3: Partly

2. Has the statistical analysis been performed appropriately and rigorously?

Reviewer #1: No

Reviewer #2: Yes

Reviewer #3: Yes

3. Have the authors made all data underlying the findings in their manuscript fully available?

Reviewer #1: Yes

Reviewer #2: Yes

Reviewer #3: Yes

4. Is the manuscript presented in an intelligible fashion and written in standard English?

Reviewer #1: Yes

Reviewer #2: Yes

Reviewer #3: Yes

Reviewer #1: This manuscript investigated the association between IL-13, age, platelet count, and liver stiffness in patients with chronic hepatitis B, and developed a predictive model which may help identify patients at risk for liver fibrosis and guide monitoring and early intervention. The study is well-structured: inclusion/exclusion criteria were clearly stated, a reasonable sample size was included, all the variables were clearly defined. The authors applied appropriate statistical methods, including transformation of liver stiffness to achieve approximate normality, checking multicollinearity to avoid bias the regression coefficients, and multiple linear regression modeling. Overall, the study provides useful insights, and the manuscript is clearly written. However, several aspects about rationale and model could be strengthened to improve clinical relevance, model robustness, and generalizability. Please see my comments below.

Introduction:

The rationale for selecting IL-13, age, and platelet count is stated; however, it is unclear why only these three factors were chosen. For example, why was ALT not considered as a potential predictor?

And the rationale for IL-13 should be stronger. The connection between IL-13 and chronic hepatitis B is not well-established in the introduction. As mentioned in the discussion, IL-13 was unrelated to LSM outcome scores based on other studies. Better explanation needed.

Methods:

Why was the IL-13 variable made into categorical data? This may lead to loss of statistical power and information; effect estimate could be less precise.

And the cutoff value was decided “based on the cut-off value previously obtained through ROC Curve analysis and the Youden index calculation.” Citation for previous data? Is it clinically meaningful? Cut-off value may not be generalized to other cohorts.

Results:

Under Table 3, the authors mentioned “The highest correlation between variables did not exceed 0.7, indirectly indicating no multicollinearity if further analysis was conducted.” But I cannot locate the data of the correlation between each variable.

In the multiple linear regression model, I recommend considering the inclusion of interaction terms, for example, Age x IL-13. Even when multicollinearity is low, interaction terms can reveal important non-additive relationships that may better reflect biological processes. For example, older age alone may only cause moderate increase in liver stiffness, and higher IL-13 also cause moderate increase in stiffness. But older patients with high IL-13 could lead to an extra-large increase. Without an interaction term, the model may oversimplify biology.

Discussion:

I would recommend discussing more about interpreting and clinical relevance of this model. And how does it compare to existing predictive markers/models like FIB-4 since it is mentioned in the introduction?

And deeper discussion on the findings in IL13 is needed. Why do the results differ from previous studies? Are there potential biological, cohort-specific, or methodological explanations?

Reviewer #2: This manuscript aimed to investigate the role of IL-13 in liver fibrosis in patients with CHB and to construct a prediction model for liver fibrosis that incorporates IL-13, age, and platelet count, it presents an interesting and relevant study on a predictive model for liver fibrosis in CHB patients.

1. A sample size of n=88 is very small for developing a predictive model. Since the model was built and evaluated only on the same dataset, its performance (R²=0.37) is likely an overestimate, and its generalization ability is questionable. Do the authors plan to conduct external validation in a larger sample or independent cohort?

2. If not, the authors should validate the model's performance in an independent cohort of CHB patients who were not involved in model construction.

3. An R² of 0.37 means the model only explains 37% of the variation in liver stiffness, with more than 60% explained by other unknown factors. This limits its clinical usefulness. Do the authors consider incorporating other variables like ALT, AST?

4. How did the authors determine the 5 pg/mL cut-off for IL-13, and would the suthors consider validating this threshold using other patient cohorts or clinical outcomes?

5. The mechanism of action of IL-13 in liver fibrosis is not fully understood, especially in the setting of CHB. Could you further elucidate the potential pathophysiology of IL-13 in CHB liver fibrosis? Are there plans to conduct in vitro or in vivo experiments to validate its role?

6. While the study addresses age, platelet count, and IL-13 levels, other potential confounders or factors influencing liver stiffness (e.g., comorbid conditions or medications) are not discussed in-depth. A brief explain should be included.

Minor concern:

1. Predictors and other parts: In this manuscript, “x 10⁹/L” is used to represent platelet count, but the “x” here should be clearly defined as the multiplication symbol (×), not the letter “x”.

Reviewer #3: The study explored the association between serum interleukin-13 (IL-13) and liver fibrosis in chronic hepatitis B (CHB) patients and developed a multiple linear regression (MLR)–based predictive model integrating age, platelet count, and IL-13, based on data from 88 CHB patients. The inverse of liver stiffness (inverseLSM=1/(LSM+1)) was used as the model outcome and the final model covered 37% of the variance in the inverseLSM results. This work is clinically relevant and it proposes an approach to address the need for non-invasive tools for liver fibrosis assessment, especially in resource-limited settings where transient elastography is not available. However, the robustness of this model and its translational value are limited by the methodological simplicity, limited sample size, and lack of validation.

Major revision:

1. IL-13’s role in liver fibrosis has also been shown in other chronic liver diseases, while its independent correlation with CHB has not been well discussed in the introduction. A more detailed literature summary could be helpful to strengthen the rationale for including IL-13 in the prediction model and the independence of IL-13.

2. Could the authors clarify whether IL-13 remains significant when platelets and ages are controlled, highlighting its independence?

3. 88 patients is acceptable for exploratory modeling while may be underpowered for robust model development. A power analysis and/or sample-size justification could be beneficial for the study design. Additionally, the authors could consider benchmarking the new model against FIB-4 and APRI to demonstrate comparative predictive ability.

Minor revision:

1. Can the authors provide a visual representation of the model, such as a scatterplot of predicted vs. observed values?

**Do you want your identity to be public for this peer review?** For information about this choice, including consent withdrawal, please see our Privacy Policy

Reviewer #1: No

Reviewer #2: No

Reviewer #3: No

---

## [Author Response · Author response to Decision Letter 1]

17 Jan 2026

I hope this letter finds you well. I would like to express my sincere gratitude for the feedback provided by the editor and reviewer for our manuscript entitled "Unveiling the Role of Interleukin-13 in Liver Fibrosis of Chronic Hepatitis B Patients: Development of a Predictive Model" Manuscript ID: PONE-D-25-42985R1. Your insightful comments and suggestions have significantly contributed to the enhancement of the quality and clarity of our work.

In response to the reviewers' feedback, we have carefully addressed each comment and made substantial revisions to improve the overall quality of the manuscript. Enclosed is the revised manuscript and a marked-up copy highlighting the specific changes made in response to the reviewer's suggestions. These revisions address the concerns the reviewers raised and improve the manuscript's overall quality. We value the reviewers' time and effort and believe our manuscript now meets the high standards of PLOS One.

Thank you for considering our resubmission. We look forward to your feedback and the potential acceptance of our work for publication.

---

## [Decision Letter · Decision Letter 1]

30 Jan 2026

Dear Dr. Bestari,

Thank you for submitting your manuscript to PLOS ONE. After careful consideration, we feel that it has merit but does not fully meet PLOS ONE’s publication criteria as it currently stands. Therefore, we invite you to submit a revised version of the manuscript that addresses the points raised during the review process.

Please respond to reviewers' comments.

We look forward to receiving your revised manuscript.

Kind regards,

Xiaosheng Tan

Academic Editor

PLOS One

Journal Requirements:

Reviewers' comments:

Reviewer's Responses to Questions

**Comments to the Author**

Reviewer #1: (No Response)

Reviewer #2: All comments have been addressed

Reviewer #3: All comments have been addressed

2. Is the manuscript technically sound, and do the data support the conclusions?

Reviewer #1: Yes

Reviewer #2: Yes

Reviewer #3: Yes

3. Has the statistical analysis been performed appropriately and rigorously?

Reviewer #1: Yes

Reviewer #2: Yes

Reviewer #3: Yes

4. Have the authors made all data underlying the findings in their manuscript fully available?

Reviewer #1: Yes

Reviewer #2: Yes

Reviewer #3: Yes

5. Is the manuscript presented in an intelligible fashion and written in standard English?

Reviewer #1: Yes

Reviewer #2: Yes

Reviewer #3: Yes

Reviewer #1: Thanks for addressing most of my comments. The authors’ rationale for dichotomizing IL-13 is reasonable given non-linearity and instability with transformation. However, dichotomization still reduces statistical power and may produce cohort-specific cut-offs. I recommend that the authors provide additional evidence that the chosen cut-off is robust, for example, through internal validation or sensitivity analysis. Moreover, the limitation that the IL-13 cut-off may be cohort- and assay-dependent and that dichotomization may reduce statistical power and produce cohort-specific thresholds should be more clearly stated.

Although the authors note that a sample size of 88 meets the minimum criteria for model development and acknowledge the need for external validation, the limitations of a relatively small cohort should still be acknowledged. Specifically, the modest sample size may affect model stability and reduce the ability to detect interaction effects, and this should be clearly acknowledged.

Regarding interaction terms, I appreciate that potential interactions were explored. However, interaction effects often require larger sample sizes to detect. To improve transparency, I recommend that the authors include a supplementary table of the interaction model results. If the interaction term is truly non-informative, reporting these results will demonstrate that the decision to exclude it is justified rather than based solely on non-significance.

Reviewer #2: The authors have addressed the core concerns raised in my initial review The revision is improved, with greater clarity regarding the study's aims, limitations, and the need for future validation.

I suggest the authors consider incorporating the two minor suggestions above (strengthening the limitation on external validation in the text and adding a brief phrase on the exploratory nature of the IL-13 cut-off in Results) before final publication.

Reviewer #3: The authors have carefully addressed my comments and I have no further comments. I believe this manuscript is now suitable for publication.

**Do you want your identity to be public for this peer review?** For information about this choice, including consent withdrawal, please see our Privacy Policy

Reviewer #1: No

Reviewer #2: No

Reviewer #3: No

---

## [Author Response · Author response to Decision Letter 2]

14 Feb 2026

We sincerely appreciate the reviewers’ thoughtful suggestions. We have carefully evaluated the recommended publications and included those that are directly relevant to our work. In addition, we have incorporated the additional data requested by the reviewers and strengthened the discussion of the study’s limitations. We believe that all reviewer and editorial comments have now been fully addressed, and we hope that the revised manuscript is suitable for publication in PLOS One and may contribute meaningfully to the field.

---

## [Decision Letter · Decision Letter 2]

26 Feb 2026

Unveiling the Role of Interleukin-13 in Liver Fibrosis of Chronic Hepatitis B Patients: Development of a Predictive Model

PONE-D-25-42985R2

Dear Dr. Bestari,

We’re pleased to inform you that your manuscript has been judged scientifically suitable for publication and will be formally accepted for publication once it meets all outstanding technical requirements.

Kind regards,

Xiaosheng Tan

Academic Editor

PLOS One

Additional Editor Comments (optional):

Reviewers' comments:

Reviewer's Responses to Questions

**Comments to the Author**

Reviewer #1: All comments have been addressed

Reviewer #2: All comments have been addressed

2. Is the manuscript technically sound, and do the data support the conclusions?

Reviewer #1: Yes

Reviewer #2: Yes

3. Has the statistical analysis been performed appropriately and rigorously?

Reviewer #1: Yes

Reviewer #2: Yes

4. Have the authors made all data underlying the findings in their manuscript fully available?

Reviewer #1: Yes

Reviewer #2: Yes

5. Is the manuscript presented in an intelligible fashion and written in standard English?

Reviewer #1: Yes

Reviewer #2: Yes

Reviewer #1: Thanks for addressing my comments. The manuscript is appropriate for publication in its current revised form.

Reviewer #2: No futher comments. The authors have addressed the core concerns raised in my initial review The revision is improved, with greater clarity regarding the study's aims, limitations, and the need for future validation.

**Do you want your identity to be public for this peer review?** For information about this choice, including consent withdrawal, please see our Privacy Policy

Reviewer #1: No

Reviewer #2: No

---

## [Editor Report · Acceptance letter]

PONE-D-25-42985R2

PLOS One

Dear Dr. Bestari,

I'm pleased to inform you that your manuscript has been deemed suitable for publication in PLOS One. Congratulations! Your manuscript is now being handed over to our production team.

Kind regards,

on behalf of

Dr. Xiaosheng Tan

Academic Editor

PLOS One